# Sex-specificity of the *C. elegans* metabolome

Russell N. Burkhardt[1], Alexander B. Artyukhin[1,5], Erin Z. Aprison[2], Brian J. Curtis[1], Bennett W. Fox [1], Andreas H. Ludewig[1], Diana Fajardo Palomino[1], Jintao Luo[3,6], Amaresh Chaturbedi[4], Oishika Panda[1], Chester J. J. Wrobel [1], Victor Baumann [1], Douglas S. Portman [3], Siu Sylvia Lee[4], Ilya Ruvinsky [2] ✉ & Frank C. Schroeder [1] ✉

Recent studies of animal metabolism have revealed large numbers of novel metabolites that are involved in all aspects of organismal biology, but it is unclear to what extent metabolomes differ between sexes. Here, using untargeted comparative metabolomics for the analysis of wildtype animals and sex determination mutants, we show that *C. elegans* hermaphrodites and males exhibit pervasive metabolomic differences. Several hundred small molecules are produced exclusively or in much larger amounts in one sex, including a host of previously unreported metabolites that incorporate building blocks from nucleoside, carbohydrate, lipid, and amino acid metabolism. A subset of male-enriched metabolites is specifically associated with the presence of a male germline, whereas enrichment of other compounds requires a male soma. Further, we show that one of the male germline-dependent metabolites, an unusual dipeptide incorporating *N,N*-dimethyl-tryptophan, increases food consumption, reduces lifespan, and accelerates the last stage of larval development in hermaphrodites. Our results serve as a foundation for mechanistic studies of how the genetic sex of soma and germline shape the *C. elegans* metabolome and provide a blueprint for the discovery of sex-dependent metabolites in other animals.

Recent studies of metabolism and related physiological responses in humans and rodent models have revealed dramatic differences between sexes[1–5]. For example, targeted metabolomic analysis of human serum revealed significant differences for one-third of annotated metabolites[6]. However, few studies in any model system have leveraged the power of untargeted metabolomic approaches based on high-resolution mass spectrometry (HRMS) to uncover novel, unannotated metabolites associated with sex.

HRMS-based untargeted metabolomic analyses in several species have revealed vast metabolic diversity, including large numbers of metabolites whose chemical structures have not yet been determined[7,8]. Interpretation of the resulting large datasets usually relies on comparative analyses of samples from different biological conditions, which enables identifying metabolites that are significantly associated with a context of interest and thus can be prioritized for detailed chemical characterization[8–11]. HRMS-based comparative metabolomics of different sexes thus has the potential to uncover unannotated metabolites whose identification can advance mechanistic understanding of sex-specific phenotypes and complement transcriptomics and proteomics.

In the model nematode *C. elegans*, discovery-oriented metabolomic analyses[12,13] have almost exclusively focused on the predominant sex, the self-fertile hermaphrodites, which account for >99% of the populations under standard conditions[14], whereas the metabolomes of

[1]Boyce Thompson Institute and Department of Chemistry and Chemical Biology, Cornell University, Ithaca, NY 14853, USA. [2]Department of Molecular Biosciences, Northwestern University, Evanston, IL 60208, USA. [3]Department of Biomedical Genetics, University of Rochester, Rochester, NY 14642, USA. [4]Department of Molecular Biology and Genetics, Cornell University, Ithaca, NY 14853, USA. [5]Present address: Chemistry Department, College of Environmental Science and Forestry, State University of New York, Syracuse, NY 13210, USA. [6]Present address: School of Life Sciences, Xiamen University, 361102 Xiamen, Fujian, China. ✉e-mail: ilya.ruvinsky@northwestern.edu; fs31@cornell.edu

the much less abundant males have been studied only to a limited extent. Males and hermaphrodites differ considerably – over 5500 genes, or ~1/3 of the entire protein-encoding transcriptome, are differentially expressed between the two sexes[15,16]. Correspondingly, there is growing evidence for major sex-specific differences in metabolism, disease response, and other phenotypes in *C. elegans*[17–21].

An intriguing subset of metabolomic differences between the sexes is comprised of excreted small molecules with which animals communicate. Several recent studies describe diverse life history traits affected by such sex-specific pheromones. The best-studied case involves a pair of ascaroside derivatives that have nearly identical chemical structures (Fig. 1a) but are enriched either in hermaphrodites (ascr#3, **1**) or in males (ascr#10, **2**)[22]. At physiological concentrations, the male-specific ascr#10 exerts effects that appear opposite to the effects of ascr#3[23], including reduced exploratory behavior of hermaphrodites[24,25], improved aspects of germline function[26–28], and

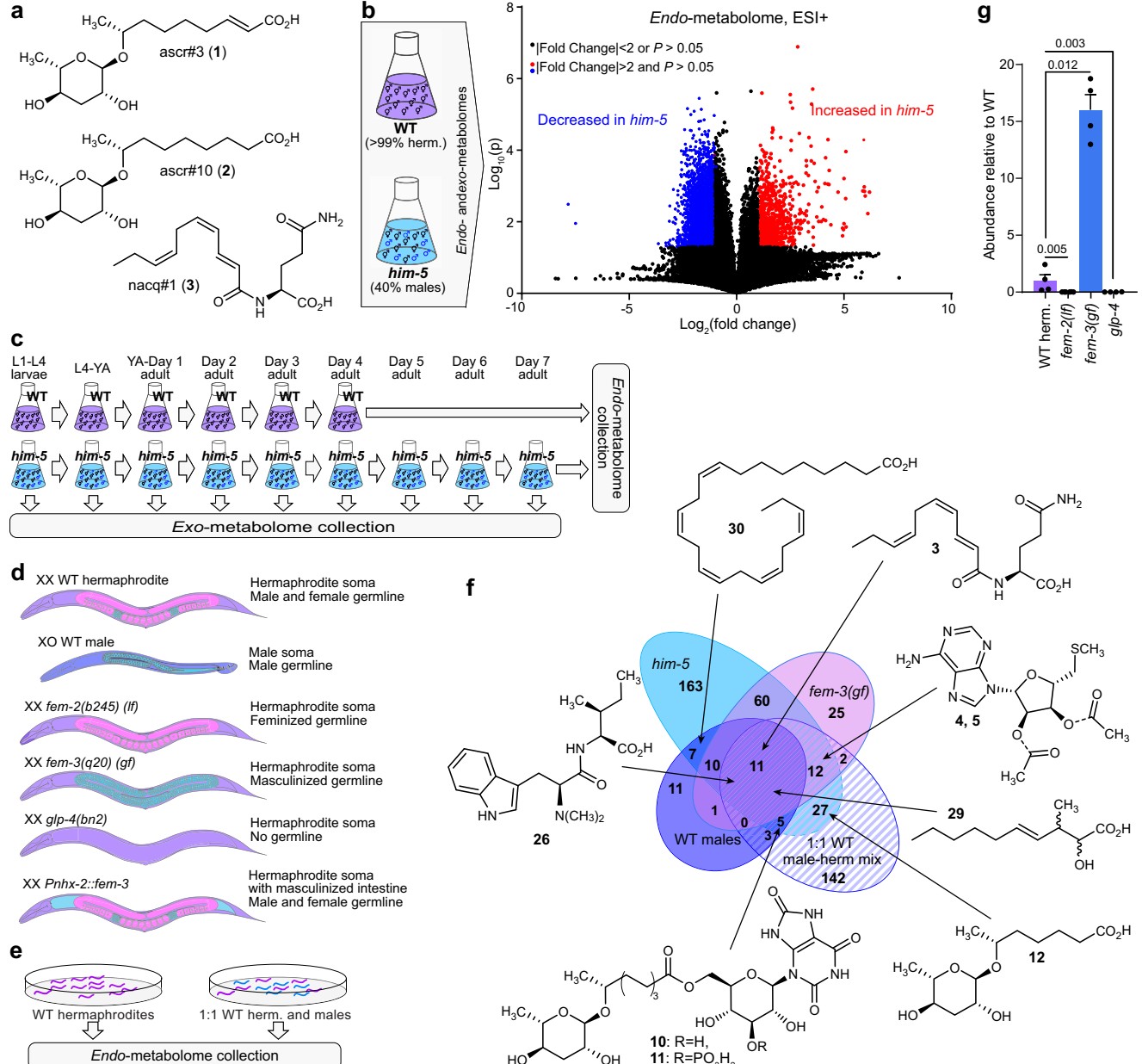

**Fig. 1 | Metabolomic profiling of male *C. elegans*. a** Previously characterized sex-biased metabolites in *C. elegans*. Whereas ascr#3 (**1**) is more abundant in hermaphrodites, ascr#10 (**2**) and nacq#1 (**3**) are more abundant in males.
**b** Comparison of the metabolomes of WT and male-enriched *him-5(e1490)* mutants. The volcano plot shows the subset of features detected by HPLC-HRMS in positive ionization mode (ESI+), highlighting features increased (red) or decreased (blue) in *him-5* relative to WT *endo*-metabolome extracts. *P* values were calculated via unpaired two-sided *t* tests; see "Metaboseek analysis", for details.
**c** Experimental setup for monitoring metabolism of WT and *him-5* animals with periodic harvesting of the *exo*-metabolome; YA young adults. **d** Simplified body plans of WT hermaphrodites and WT males as well as germline-feminized *fem-2(b245) (lf)*, germline-masculinized *fem-3(q20) (gf)*, germline-deficient *glp-4(bn2)*,

and intestinally masculinized *Pnhx-2::fem-3* mutants. Tissues are color-coded according to sex; hermaphrodite soma (dark purple), male soma (dark blue); female germline pink), male germline (blue) with sperm (olive dots), and male intestine (light blue). Reprinted from ref. [61] with permission from Elsevier.
**e** Schematic of hermaphrodite and male-enriched WT animals grown on plates for metabolomics. **f** Venn Diagram of male-enriched metabolites detected from *him-5* mutants, *fem-3(gf)* mutants, 1:1 mixtures of WT males and hermaphrodites, and hand-picked WT males. **g** Abundance of nacq#1 (**3**) in *exo*-metabolome extracts of germline mutants relative to WT hermaphrodites. Data represent four or six biologically independent experiments, and error bars represent mean ± s.e.m. *P* values were calculated using the $\log_{10}$-transformed raw data and two-sided Welch *t* tests. Source data are provided as a Source Data file.

shortened lifespan[27,29]. More recently, we reported an unusual fatty acid-amino acid conjugate, nacq#1 (**3**), that also shortens the hermaphrodite lifespan[29], therefore contributing to a phenomenon of male-induced demise, whereby hermaphrodites die more quickly in the presence of male-excreted compounds[19,30,31]. In addition to reducing hermaphrodite lifespan, nacq#1 accelerates the last stage of larval development, resulting in faster sexual maturation[29].

Studies of ascr#10 and nacq#1 demonstrated that sex-specific metabolites can have major effects on *C. elegan's* life history and suggest that the identification of additional sex-specific compounds will contribute significantly to a mechanistic understanding of various aspects of *C. elegans* biology. Here, we report a comprehensive untargeted survey of differences between male and hermaphrodite metabolomes, including the roles of male and female germlines in the observed differences. The metabolomic data from this study provide a resource for future detailed exploration of sex-biased metabolic pathways and phenotypes.

## Results

### Approaches toward sex-specific metabolomics

One reason for the relatively scant knowledge of the *C. elegans* male metabolome is the low abundance of males in the wild-type (WT) laboratory strain, N2 Bristol. Under standard laboratory conditions, WT cultures consist almost entirely of self-fertilizing hermaphrodites that carry two X chromosomes, whereas males with a single X chromosome constitute <1% of the population[32]. Although mating between hermaphrodites and males can yield up to 50% male progeny, this form of reproduction occurs primarily on plates and less effectively in liquid culture. The low male frequency and the requirement for plates for mating make it impractical to prepare large samples of animals as needed for in-depth metabolomic analyses.

Therefore, we began our analyses with a comparison of the metabolomes of WT cultures, consisting mostly of hermaphrodites, and *him-5(e1490)* mutant cultures, which contain up to 40% males (Fig. 1b). The *him-5(e1490)* animals carry a mutation that increases the likelihood of nondisjunction of X chromosomes during meiosis, resulting in a higher incidence of males, independent of mating[32]. To distinguish metabolites that are primarily excreted, we separately analyzed extracts of the culture media (*exo*-metabolome) and the worm bodies (*endo*-metabolome). Next, we considered that metabolism undergoes profound changes during development and throughout adult life. Elucidation of biological functions and biosynthesis of sex-specific compounds could be aided by the knowledge of the developmental stages during which they are produced. To this end, we examined the *exo*-metabolome of *him-5* cultures by collecting the culture media during different time periods throughout larval development and adulthood (Fig. 1c).

Metabolome samples were analyzed by liquid chromatography coupled to a high-resolution mass spectrometer (LC-HRMS). Bioinformatic analysis of the resulting data was conducted using the Metaboseek platform, which provides a comprehensive toolset for comparative metabolomics (see "Methods"[8]). Our analysis of the *exo*- and *endo*-metabolomes of WT and *him-5* cultures yielded 110,184 significant features (unique pairs of mass-to-charge (*m/z*) ratios and retention times), including 8225 features at least twofold more abundant in the male-enriched *him-5* cultures and 22,538 MS features that are at least twofold more abundant in WT cultures (Fig. 1b). Manual curation to remove isotopes, adducts, and fragments yielded almost 300 unique compounds that are upregulated in male-enriched *him-5* cultures, and a similar number of compounds enriched in WT relative to *him-5*. Following a comparative analysis of the LC-HRMS data, we acquired MS2 fragmentation spectra for all male-enriched metabolites (Supplementary Data 1–3).

### Prioritization of sex-dependent compounds

Both the germline and somatic tissues differ between the two sexes and could therefore contribute to the detected differences between *him-5* and WT animals. The self-fertile *C. elegans* hermaphrodites produce a cache of ~300 sperm, several-fold fewer than males, before switching to oogenesis[33]. To begin disentangling potential sources of metabolic differences, we took advantage of three conditional mutations that at the non-permissive temperature alter aspects of germline biology (Fig. 1d). The XX individuals (these are normally hermaphrodites) carrying the *fem-3(q20)* gain-of-function (*gf*) allele have a masculinized germline that produces a dramatically increased amount of sperm and no eggs, but do not show overt masculinization in the soma[34]. Conversely, the XX individuals carrying the *fem-2(b245)* loss-of-function (*lf*) allele have a feminized germline that produces eggs but no sperm[35]. The third mutation we used in this study, *glp-4(bn2)*, severely limits germline development (~1% of the wild-type number of germline nuclei), while maintaining an apparently normal soma[36], although a likely null allele causes larval lethality[37]. In addition to analyzing large liquid cultures of the three mutant strains (*fem-3*, *fem-2*, and *glp-4*), we analyzed small, hand-picked, plate-derived samples, that were too small to permit extensive metabolomic analyses, but sufficient to verify our results. In this way, we confirmed that many of the compounds enriched in *him-5* and *fem-3(gf)* mutants were enriched in WT males and in a 1:1 mixture of WT males (derived by heat shock) and hermaphrodites (Fig. 1e). We also used hand-picked samples to further narrow down somatic tissues likely involved in the production of sex-biased small molecules. Previous metabolomic analyses of *C. elegans* identified the intestine as a major somatic source of small molecule biosynthesis[11]. We therefore analyzed the production of male-enriched compounds in hermaphrodites with masculinized intestines, focusing on molecules whose abundances were not significantly altered in germline mutants. Intestinally masculinized worms were obtained by expressing the male sexual regulator FEM-3 under control of the promoter of the *nhx-2* gene (*Pnhx-2::fem-3*)[38,39].

Comparison of metabolomes of *him-5* animals and germline-masculinized *fem-3(gf)* animals revealed that 93 of the 295 *him-5*-enriched compounds were also upregulated in *fem-3(gf)* cultures (Fig. 1f), suggesting that production of this subset of metabolites is associated with the presence of the male germline, whereas production of the remaining 202 *him-5*-enriched compounds may depend on the male soma or other changes in metabolism in *him-5* mutants. The majority of metabolites enriched in *fem-3(gf)* hermaphrodites relative to WT were also enriched relative to germline-feminized *fem-2(lf)* and germline-less *glp-4* mutants, further supporting that these metabolites depend on the presence of the male germline (Supplementary Data 1–3). An example of a *him-5*-enriched metabolite that depends on the presence of the male germline is the developmental regulator nacq#1, which was previously shown to be produced in much higher amounts by males relative to hermaphrodites[29]. Quantification of nacq#1 in the germline mutants showed that this metabolite was abundantly produced in masculinized *fem-3(gf)* animals, but absent in feminized *fem-2(lf)* and germline-less *glp-4* hermaphrodites (Fig. 1g). nacq#1 abundance was unchanged in the intestinally masculinized *Pnhx-2::fem-3* hermaphrodites (Supplementary Fig. S1a). Comparison of the metabolomes of *fem-3(gf)*, *him-5*, and WT cultures revealed 11 additional *him-5*-enriched compounds that were missed in the initial comparison of the *him-5* and WT metabolomes (Supplementary Data 1–3).

For chemical characterization of male-enriched metabolites, we initially prioritized compounds associated with the presence of the male germline, that is, increased in *him-5* as well as *fem-3(gf)* compared to WT cultures. Analysis of the molecular formulae and MS2 networks

for the resulting set of compounds revealed several different families of likely structurally related metabolites. For in-depth analysis, we then selected representative and most abundant members of these families that could also be detected in samples of WT males.

## Male-enriched nucleoside derivatives

Among prioritized male-enriched metabolites in the *endo*-metabolomes, we detected several families of compounds whose molecular formulae and MS2 fragmentation patterns suggested that they represent unusual nucleoside derivatives. These included two isomeric compounds ($m/z$ 340.1074, $C_{13}H_{18}N_5SO_4^+$, 6.69, and 7.30 min) that fragmented in a nearly identical manner (Fig. 2a), with the later-eluting compound being the predominant isomer. MS2 fragments consistent with methylmethylenesulfide ($m/z$ 61.0104, $C_2H_5S^+$) and adenine ($m/z$ 136.0608, $C_5H_6N_5^+$) suggested that the two compounds represent *O*-acetylated derivatives of *S*-methylthioadenosine (MTA), a common metabolite downstream of *S*-adenosylmethionine (SAM)[40]. Acetylation of MTA yielded a mixture of two isomers of the same retention time as the natural compounds (acemta#1, **4**, and acemta#2, **5**), confirming their structures (Supplementary Figs. 2 and 3).

Production of acemta#1 and acemta#2 appears to be associated with the presence of the male germline, as the two compounds are undetectable in feminized *fem-2(lf)* and germline-less *glp-4* animals, whereas they are much enriched in masculinized *fem-3(gf)* animals compared to WT hermaphrodites (Fig. 2b). acemta#1 and acemta#2 were also enriched in samples of hand-picked males (Fig. 2c). In contrast to ascaroside pheromones and nacq#1[22,29], these nucleoside derivatives accumulate primarily in the worm body, and their abundance is increased in older worms (Fig. 2d). The abundance of the putative precursor of acemta#1/2, MTA, is not significantly increased in *him-5* or *fem-3(gf)* mutants (Supplementary Fig. 4).

In addition to acemta#1 and acemta#2, we detected a family of compounds whose MS2 spectra suggested that they represent hexose-based nucleosides, including a series of putative uric acid derivatives. Molecular formulae and MS2 spectra of the two most dramatically enriched (75- and 23-fold over WT in the *him-5 endo*-metabolome) members of this family ($m/z$ 587.2211, $C_{24}H_{36}N_4O_{13}$ and $m/z$ 667.1880, $C_{24}H_{37}N_4PO_{16}$) suggested that they represent isomers of the recently described uglas#1 (**6**) and uglas#11 (**7**), two gluconucleosides incorporating uric acid and the ascaroside ascr#1 (Fig. 2e, f)[41]. However, uglas#1 and uglas#11, which bear an ascr#1 moiety in the 2-position of the glucose, had significantly earlier retention times than their male-enriched isomers and were virtually absent in *him-5* mutants. Based on the recent identification of a series of 6'-*O*-acylated glucosides from *C. elegans*[42,43], we hypothesized that in the male-enriched isomers of uglas#1 and uglas#11 the ascaroside moiety is attached to the 6-position of the uric acid glucoside. Comparison of MS2 spectra and retention times of a synthetic sample of the 6'-*O*-substituted isomer of uglas#1, named uglas#14 (**8**), established that the male-enriched isomer is in fact uglas#14, and thus the male-enriched isomer of uglas#11 was assigned as its phosphorylated derivative, uglas#15 (**9**, Fig. 2f and Supplementary Figs. 5 and 6). uglas#14 and uglas#15 were accompanied by two compounds with analogous fragmentation patterns (Supplementary Figs. 7 and 8), which appeared to incorporate the 9-carbon sidechain ascaroside, ascr#10, instead of the 7-carbon sidechain ascr#1. These compounds, named uglas#104 ($m/z$ 615.2531, $C_{26}H_{40}N_4O_{13}$, **10**) and uglas#105 ($m/z$ 695.2189, $C_{26}H_{41}N_4PO_{16}$, **11**), were roughly three- to four-fold enriched in *him-5 endo*-metabolome compared to WT cultures, whereas uglas#14 and uglas#15 were enriched 20-60-fold (Fig. 2g). Biosynthesis of the 2'-*O*-acylated uglas#1 and uglas#11 has been shown to require the carboxylesterase *cest-1.1*, which mediates attachment of the ascaroside to the 2'-hydroxy of uric acid gluconucleosides[42]. Production of the male-upregulated

6'-*O*-acylated uglas-family metabolites is not *cest-1.1*-dependent and does not require any of the other so-far characterized *cest* homologs[42,43], suggesting that a different carboxylesterase is involved.

All four male-enriched uglas#-family metabolites that were enriched in the *endo*-metabolomes of large *him-5* cultures were also enriched in small hand-picked samples of N2 and *him-5* males (Fig. 2h). As was the case with the larger *him-5* cultures, production of the 2'-*O*-acylated uglas#11 was significantly decreased or even abolished in hand-picked samples of males relative to WT hermaphrodites, whereas abundances of the 6'-*O*-acylated isomer, uglas#15, and of uglas#104/105 were dramatically increased (Fig. 2h). Similar to acemta#1 and acemta#2, the uglas#-family metabolites were predominantly retained within the worm body, and thus our experimental setup for determining changes of abundances during the adult lifespan (see Fig. 1c) allowed only for comparison of abundances in young adults and 6-day-old adults, which indicated a modest increase of uglas#104 and uglas#105 in day-6 adult worms (Supplementary Fig. 9).

In contrast to other male-enriched metabolites, including nacq#1 and acemta#1/2, production of the male-enriched uglas#-family metabolites was not strictly dependent on the presence of a male (or female) germline. Abundances of the male-specific uglas#-family metabolites were not reduced in feminized *fem-2(lf)* animals relative to WT, only slightly increased in masculinized *fem-3(gf)* animals, and either slightly enriched or not changed in germline-less *glp-4* animals relative to WT (Fig. 2i). This suggested that production of uglas#-family metabolites primarily depends on the sex of somatic tissues. In fact, we found that the profile of uglas#-family metabolites in the *endo*-metabolome of intestinally masculinized *Pnhx-2::fem-3* hermaphrodites closely resembles that of males (Fig. 2j).

## Male-enriched ascaroside derivatives

Several recent studies have shown that the ascaroside ascr#10, the first reported male-enriched *C. elegans* metabolite[22], has broad effects on hermaphrodite development and physiology[24–29]. The original work reporting increased levels of ascr#10 in males was based on the analysis of mixed-stage *him-5* cultures, containing animals at different larval stages and ages of adulthood. In these *him-5* cultures, ascr#10 was found to be upregulated by approximately threefold[22]. Our *exo*-metabolome survey throughout larval development and adulthood revealed that only trace amounts of ascr#10 are produced during the four larval stages (L1-L4), and that production increases dramatically in young adults. In *him-5* cultures, secretion of ascr#10 increased until day 4 of adulthood and was roughly tenfold higher than in WT cultures starting in young adults and throughout the course of the experiment (Fig. 3a). ascr#10 production was largely unchanged from WT levels in the three tested germline mutants, but greatly increased in intestinally masculinized *Pnhx-2::fem-3* hermaphrodites (Fig. 3b). These animals also exhibited a large decrease in the production of ascr#3 (Supplementary Fig. 1c), whose synthesis is hermaphrodite-biased[22]. These results are consistent with previous studies that showed that the bulk of secreted ascarosides are derived from intestinal biosynthesis[11] and further indicate that the sexual state of the intestine is sufficient to implement sex-typical patterns of the production of these pheromones.

In addition to ascr#10, our comparative analysis of *him-5* and WT cultures revealed significantly increased amounts of the ascaroside ascr#1 (**12**) (Supplementary Fig. 10) and a series of structurally more complex ascaroside derivatives, primarily in the *him-5 endo*-metabolome (Fig. 3c, d). These included the phosphorylated phascr#11 (**13**) and the previously described glucoside glas#1 (**14**)[44,45] which were identified based on analysis of the MS2 spectra (Supplementary Figs. 11 and 12) and comparison of retention times. MS2 spectra further allowed proposing structures for several additional compounds (Supplementary Figs. 13–17), including a phosphorylated derivative glas#11 (**15**), the previously described conjugates of ascr#1

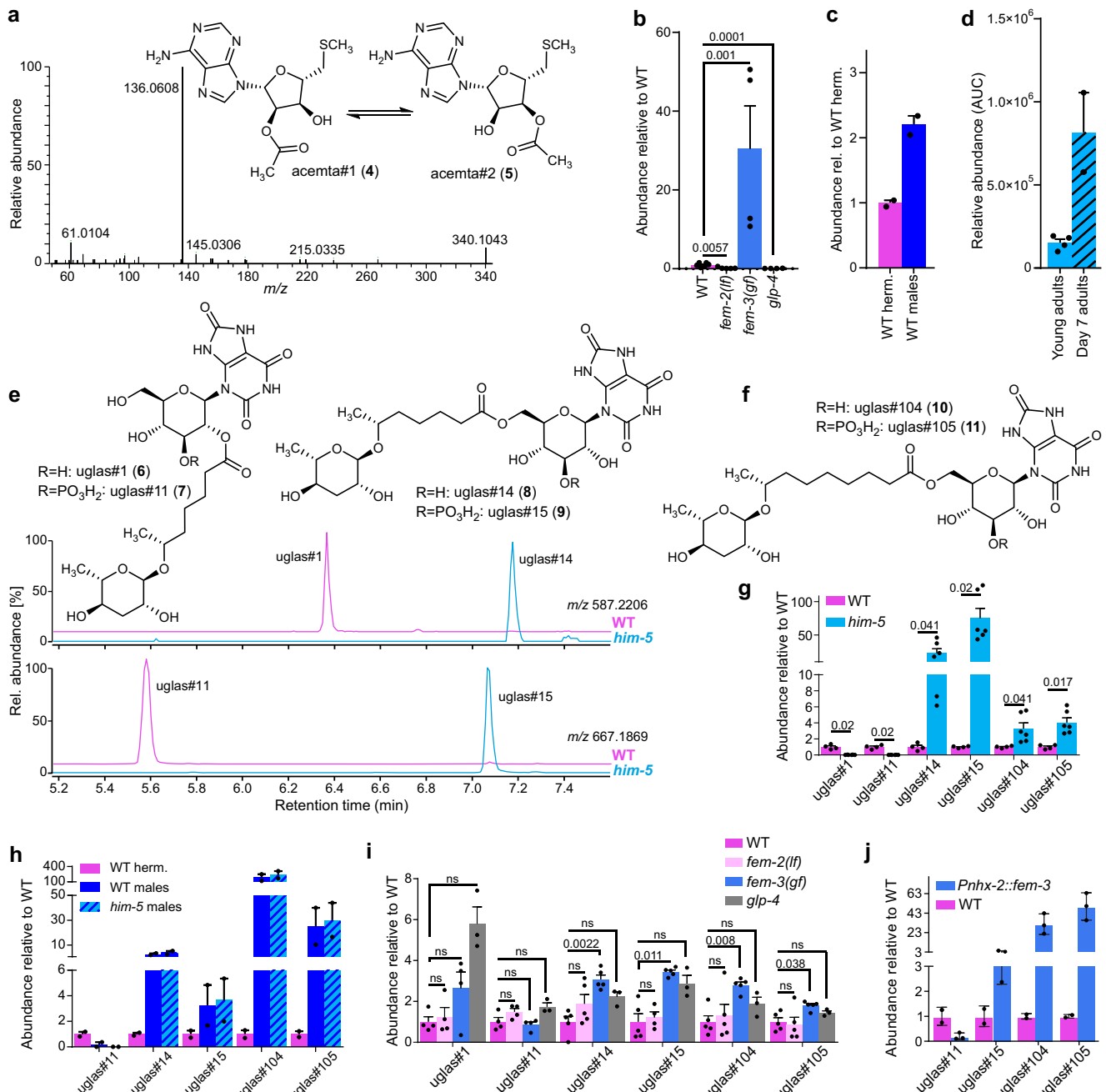

**Fig. 2 | Nucleoside-derived metabolites are enriched in males. a** MS2 fragmentation of acemta#2 (**5**) in positive (ESI +) ionization mode. **b** Abundance of acemta#2 in *endo*-metabolome extracts of *fem-2(lf)*, *fem-3(gf)*, and *glp-4* mutants relative to WT. Data represent four or six biologically independent experiments, and error bars represent mean ± s.e.m. *P* values were calculated using the log10-transformed raw data and two-sided Welch *t* tests. **c** Comparison of acemta#2 levels in hand-picked WT males and hand-picked WT hermaphrodites. Data represent two biologically independent experiments, and error bars represent mean ± s.e.m. **d** Comparison of acemta#2 levels in young adults and old (day 7) *him-5* adults. Data represent two (day-7 adults) and four (young adults) biologically independent experiments, and error bars represent mean ± s.e.m. **e** Structures and sex-specificity of uric acid glucoside derivatives incorporating the 7-carbon sidechain

ascaroside, ascr#1 (**12**). Shown are ion chromatograms obtained in ESI- ionization mode of WT and *him-5 endo*-metabolome samples. **f** Additional male-enriched uric acid glucoside derivatives, uglas#104 (**10**) and uglas#105 (**11**), incorporating a 9-carbon sidechain ascaroside, ascr#10 (**2**). **g** Abundances of uric acid glucoside-containing ascarosides (uglas#) in *endo*-metabolome extracts of (**g**) *him-5* (light blue) relative to WT (purple), **h** WT males (dark blue) and *him-5* males (blue cross-hatched) relative to WT hermaphrodites (purple), **i** germline mutants *fem-2(lf)* (light pink), *fem-3 (gf)* (blue), and *glp-4* (gray) relative to WT (purple), and **j** intestinally masculinized animals (blue) relative to WT hermaphrodites (purple). Bars in **g–j** represent mean ± s.e.m. with four to six (**g**, **i**) and two (**h**, **j**) independent biological replicates. *P* values were calculated via two-sided Welch *t* tests with Holm–Šídák correction; ns, not significant. Source data are provided as a Source Data file.

with anthranilic acid glucosides anglas#1 (**16**)[11] and anglas#2 (**17**), a putative ascr#1-derivative of tyramine glucoside[43] tyglas#1 (**18**), a putative dimeric ascaroside[46] dasc#13 (**19**), and a conjugate of ascr#1 and *p*-amino benzoic acid ascr#801 (**20**), representing a dihydro derivative of the known dauer pheromone component ascr#8 (**21**)[47].

The glucosides glas#1 and glas#11 were accompanied by later-eluting isomers with near identical MS2 spectra (Supplementary Figs. 12 and 13) that likely represent glos#1 (**22**) and glos#11 (**23**), which feature an ω-oxygenated fatty acid chain instead of the (ω−1)-oxygenated in ascr#1, glas#1, and glas#11.

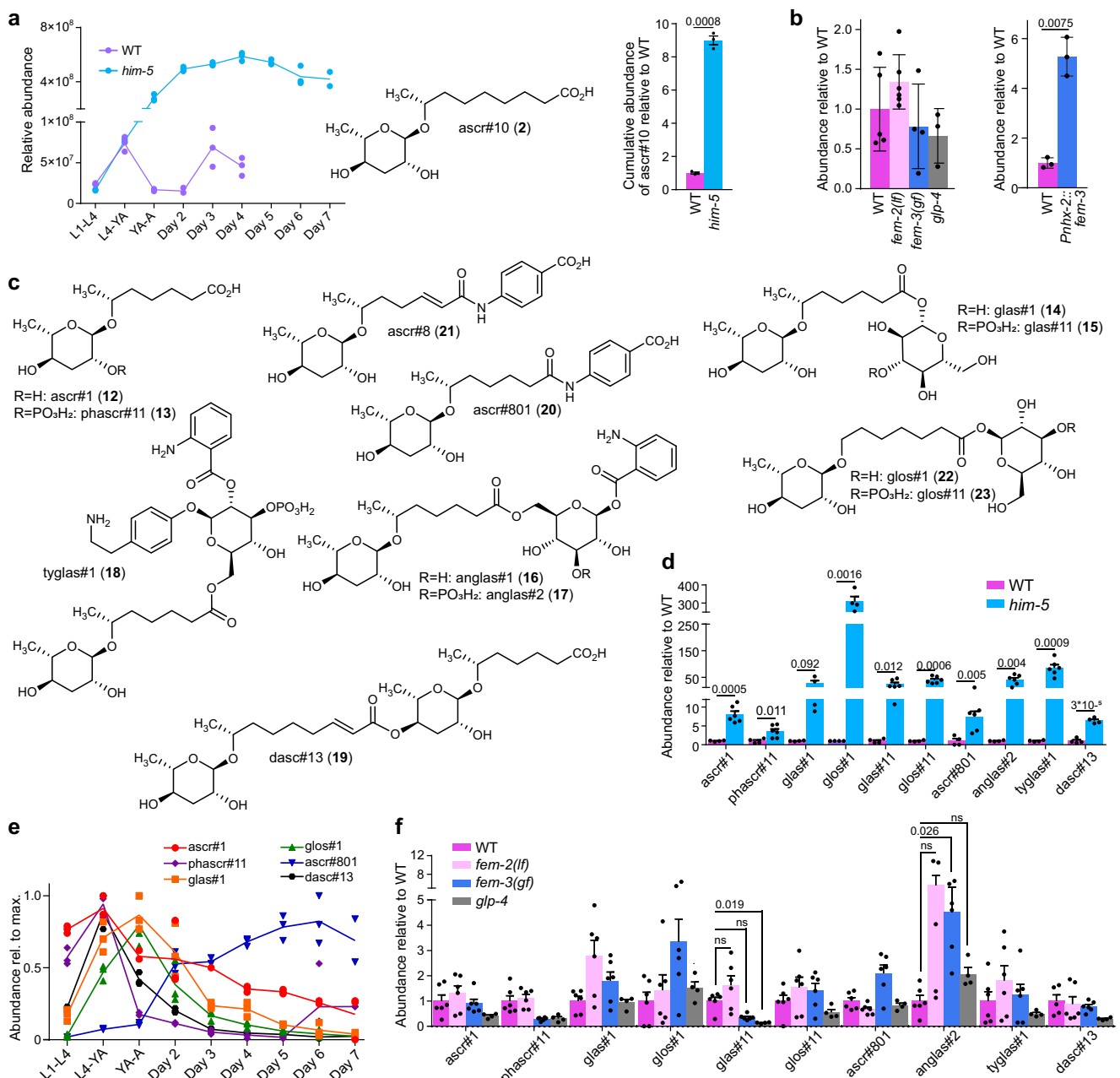

**Fig. 3 | Enrichment of ascr#10 and ascr#1 derivatives in males. a** Developmental stage-dependent production of ascr#10 (**2**) in WT (purple) and *him-5* (light blue) *exo*-metabolome extracts (left) and cumulative production of ascr#10 by WT and *him-5* from the L1 larval stage through day 4 of adulthood (right). Data represent two or three biologically independent experiments, and error bars represent mean ± s.e.m. *P* values were calculated via two-sided Welch *t* tests. **b** Abundance of ascr#10 in germline mutants (left) and intestinally masculinized *Pnhx-2::fem-3* animals (right) relative to WT. Data represent three to six biologically independent experiments, and error bars represent mean ± s.e.m. *P* values were calculated via two-sided Welch *t* tests. **c, d** Chemical structures (**c**) and relative abundances (**d**) of male-enriched C₇-sidechain ascaroside derivatives in extracts of the *exo*- (ascr#1,

phascr#11, glas#1, glos#1, ascr#801, and dasc#13) and *endo*-metabolomes (glas#11, glos#11, anglas#2, and tyglas#1) of WT (purple) and *him-5* (light blue) animals. **e** Developmental stage-dependent production of ascaroside-derived metabolites shown in (**c**) in *him-5* animals. Data represent three independent biological replicates, except for two independent biological replicates for day 7. **f** Abundances of C₇-sidechain ascaroside derivatives in extracts of germline mutants *fem-2(lf)* (light pink), *fem-3 (gf)* (blue), and *glp-4* (gray) relative to WT (purple). **d, f** Data represent four or six biologically independent experiments, and error bars represent mean ± s.e.m. *P* values were calculated by unpaired, two-sided Welch *t* test with Holm–Ší-dák correction; ns not significant. Source data are provided as a Source Data file.

Excretion of most *him-5* upregulated ascarosides, including ascr#1 itself, peaked early in life, either around the L4-to-young adult molt or the first day of adulthood (Fig. 3e). Similarly, ascr#1 derivatives primarily retained in the worm body were more abundant in young adults than in day-6 adults (Supplementary Fig. 18). In an interesting exception, ascr#801, which was barely produced early in life, was most abundant in older animals (~day 6 of adulthood). Similar to the uglas#-family nucleosides described above,

abundances of most *him-5* enriched ascr#1 derivatives did not correlate with the presence of a male germline (Fig. 3f). However, in contrast to the uglas#-family of compounds, most of the *him-5* enriched ascr#1 derivatives were not enriched or could not be detected in samples of intestinally masculinized hermaphrodites or pure males (Supplementary Fig. 19), suggesting that production of ascr#1-related metabolites may be upregulated in response to male-hermaphrodite interactions.

## An unusual male-enriched dipeptide

The comparison of the *exo-* and *endo-*metabolomes of *him-5* and WT animals revealed a small family of male-enriched metabolites whose MS2 spectra suggested they represent dipeptide derivatives. The MS2 spectrum of the most abundant member of this compound family (medip#1, *m/z* 344.1978, $C_{19}H_{26}N_3O_3^-$, 7.94 min) showed fragments suggesting the presence of putative indole (*m/z* 116.0512, $C_8H_6N^-$), and (iso)leucine (*m/z* 130.0872, $C_6H_{12}NO_2^-$) moieties, as well as fragments consistent with loss of methylindole (*m/z* 215.1413, $C_{10}H_{19}N_2O_3^-$), loss of $CO_2$ and dimethylamine (*m/z* 255.1516, $C_{16}H_{19}N_2O^-$), and loss of $CO_2$ (*m/z* 300.2102, $C_{18}H_{26}N_3O^-$) (Fig. 4a). This fragmentation pattern suggested a dipeptide consisting of *N,N*-dimethyltryptophan and either leucine or isoleucine. To confirm these assignments and to differentiate between the incorporation of leucine or isoleucine, we synthesized the two candidate compounds (Fig. 4b). Dimethylation of tryptophan with formaldehyde and sodium cyanoborohydride yielded *N,N*-dimethyltryptophan (**24**), which was then conjugated to either *O*-tBu-leucine or *O*-tBu-isoleucine, followed by deprotection with trifluoroacetic acid[10]. Comparison of retention times and MS2 spectra of isomers **25** and **26** showed that this male-enriched metabolite represents *N,N*-dimethyltryptophan-isoleucine, that we named medip#1 (**26**, Fig. 4c and Supplementary Fig. 20). medip#1 was accompanied by smaller amounts of the corresponding *N*-oxide, medip#2 (**27**, Supplementary Fig. 21), and the monomethylated derivative medip#3 (**28**).

medip#1 was enriched not only in cultures of *him-5* animals (Fig. 4d), but also in cultures with equivalent numbers of males and hermaphrodites (Fig. 4e), further confirming that this compound is enriched in males. Production of medip#1 was greatly increased in masculinized *fem-3(gf)* animals, but undetectable in feminized *fem-2(lf)* and germline-less *glp-4* animals, indicating its production is associated with the presence of a male germline (Fig. 4f). Consistent with the germline origin, medip#1 production was unchanged from WT levels in intestinally masculinized *Pnhx-2::fem-3* hermaphrodites (Supplementary Fig. 1b). medip#1 was produced abundantly in *him-5* cultures for several days after reaching the adult stage, whereas WT hermaphrodite cultures produced modest amounts of this compound and only transiently during the late larvae/young adult stage (Fig. 4g).

Feeding of methyl-$D_3$ methionine to *C. elegans* resulted in robust $D_3$- and $D_6$-labeling of medip#1–3 (Supplementary Fig. 22), indicating that methylation of the tryptophan moiety in these dipeptides proceeds via an *S*-adenosylmethionine- (SAM-)dependent methyltransferase. Notably, the free amino acid *N,N*-dimethyltryptophan (**24**), which we detected in *exo-* and *endo-*metabolome samples of WT and *him-5* mutants, remained unlabeled in the $D_3$-methionine feeding experiment (Supplementary Fig. 23). In contrast to medip#1–3, dimethyltryptophan was also detected in extracts of the *E. coli* OP50 bacteria used as food, suggesting that the detected dimethyltryptophan is of bacterial origin. These results suggest that medip#1–3 are derived from the methylation of a precursor peptide in *C. elegans*, rather than from a peptide-forming reaction using *N,N*-dimethyltryptophan. Finally, we asked whether abundances of non-methylated dipeptides of tryptophan with isoleucine or leucine are also affected by the presence or absence of a male germline. They were not—these dipeptides were similarly abundant in *fem-3(gf)* animals (these produce large amounts of medip#1) and *fem-2(lf)* animals (undetectable medip#1) as in WT hermaphrodites (Supplementary Fig. 1d).

## Conservation and biological activity of medip#1

Next, we asked whether male-enriched production of the metabolites identified in the preceding sections is conserved in other nematodes. A commonly used reference point for comparisons with *C. elegans* is *C. briggsae*, a member of the same genus that also typically reproduces by self-fertilization. Similar to *C. elegans him-5*, cultures of *C. briggsae CBR-him-8* mutants produce increased numbers of males[48] (up to 30% in our hands). Comparing the *exo-* and *endo-*metabolomes of

*C. briggsae CBR-him-8* cultures and the *C. briggsae* WT strain AF16 revealed a similar number of different features as in our comparison of *C. elegans* WT (N2) and *CEL-him-5* metabolomes. However, most male-enriched metabolites we identified in *C. elegans* could not be detected or were not enriched in *C. briggsae* males, consistent with other recent studies demonstrating that the metabolomes of nematodes are highly species-specific[13]. Notable exceptions include the previously reported nacq#1[29] as well as the dipeptide medip#1 (**26**, Fig. 4h), which was enriched in the *CBR-him-8 exo-*metabolome to a similar extent as in *CEL-him-5*.

Since male-enriched production of medip#1 is conserved, we selected this compound for further biological evaluation. medip#1 is much more abundant in the *exo-* than in the *endo-*metabolomes in both *C. elegans* and *C. briggsae*, indicating that it is preferentially excreted by males and may serve as a chemical signal, potentially eliciting responses in hermaphrodites. We found that hermaphrodite larvae reached morphologically-defined adulthood faster on plates conditioned with synthetic medip#1 at concentrations that are similar to those in the analyzed *exo-*metabolome samples (Supplementary Fig. 24a). This acceleration is due to shortening of the last larval stage (L4) because the developmental progression of earlier larval stages is not affected by medip#1 (Supplementary Fig. 24b). Faster development of somatic tissues was accompanied by a faster maturation of the oogenic germline as medip#1-exposed hermaphrodites acquired oocytes faster (Fig. 4i) and started to ovulate earlier (Fig. 4j). Development of the soma and the germline are energetically demanding. Plausibly to sustain accelerated sexual maturation, medip#1 increases the rate of food consumption (Fig. 4k). The result of the totality of physiological changes induced by medip#1 in hermaphrodites is the earlier onset of reproduction (Fig. 4l), which may confer a competitive advantage under specific conditions during the boom-and-bust cycles characteristic of the ephemeral environments where *C. elegans* dwell[14]. However, faster sexual maturation appears to come at the cost of a shortened adult lifespan. Exposure to nanomolar concentrations of medip#1 reduces lifespan by up to 12%, similar to the effects of nacq#1 (Fig. 4m).

Given the physiological effects of medip#1, we asked whether dipeptides that are chemically closely related to medip#1 have similar activities. To test this, we selected tryptophyl-isoleucine (Trp-Ile), corresponding to a putative non-methylated precursor of medip#1. Trp-Ile did not accelerate development, induce earlier onset of reproduction, or increase pharyngeal pumping rate (Supplementary Fig. 25a–d). We additionally tested the reverse peptide, isoleucyl-tryptophan (Ile-Trp) and a chemically unrelated male-enriched metabolite, bemeth#2, for effects on the onset of reproduction. Neither Ile-Trp nor bemeth#2 showed any activity (Supplementary Fig. 25d–f). This structure-specificity of the activity of medip#1 is similar to the case of previously described nacq#1, which also accelerates development, whereas its closely related structural isomer, nacq#2, is inactive[29]. Lastly, because the physiological effects of medip#1 are overall similar to those of the chemically unrelated nacq#1[29], we tested whether developmental acceleration by medip#1 and nacq#1 are additive or perhaps synergistic. However, the combination of medip#1 and nacq#1 was no more effective than either compound alone at the assayed concentration (Supplementary Fig. 26). Taken together, these results indicate that specific male-secreted metabolites, including medip#1 and nacq#1, regulate hermaphrodite development and food consumption, whereas chemically closely related compounds present in the *C. elegans exo-*metabolome and other male-enriched metabolites do not affect these phenotypes.

## Discussion

Our comparative metabolomic analyses of *C. elegans* males and hermaphrodites revealed several hundred significantly male-enriched metabolites, for which we provide MS and MS2 data as a basis for future studies toward their detailed biochemical and functional

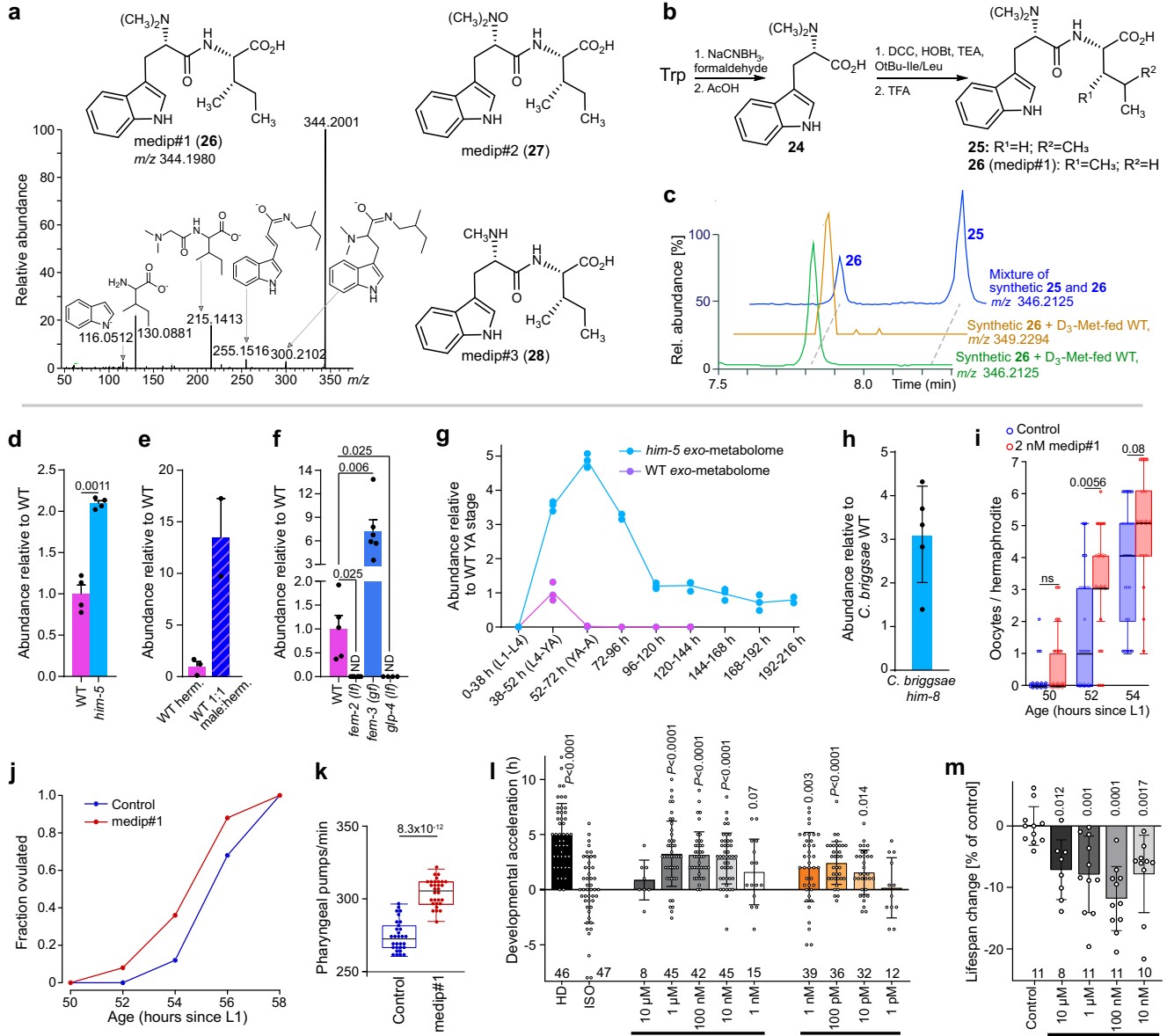

**Fig. 4 | Identification, germline dependence, and biological activity of dipeptide medip#1. a** Chemical structures of medip#1–3 (**26–28**) and MS2 fragmentation of medip#1 (**26**) in ESI‑ mode. **b** Synthesis of medip#1 and related molecules. **c** Ion chromatograms for *m/z* 346.2125 and 349.2294, corresponding to medip#1 and D₃‑medip#1, from *exo*‑metabolome extracts of *him‑5* animals supplemented with D₃‑Met and a synthetic sample containing medip#1 (**26**) and its Leu‑derived isomer (**25**). **d, f** Abundance of medip#1 in *exo*‑metabolome extracts of (**d**) *him‑5* relative to WT, **e** 1:1 male:hermaphrodite mixtures relative to WT hermaphrodites, and **f** indicated germline mutants relative to WT. Data represent four (**d**), two (**e**), and four or six (**f**) biologically independent experiments, and error bars represent mean ± s.e.m. *P* values were calculated by two‑sided Welch *t* tests; ND, not detected. **g** Developmental stage‑dependent production of medip#1 in WT (purple) and *him‑5* (light blue) animals. Data are from three independent biological replicates, except for two independent biological replicates for day 7. **h** Abundance of medip#1 in the *exo*‑metabolome of male‑enriched *C. briggsae him‑8* mutants relative to WT *C. briggsae*. Data represent five biologically independent experiments. **i** Faster acquisition of oocytes in hermaphrodites exposed to 2 nM of medip#1 (red) compared to paired controls (blue). Boxes represent the two inner quartiles, horizontal lines represent medians, and whiskers extend to 1.5× of the box data. Data represent one experiment with *n* = 25 animals for each condition; *P* values were calculated by Kolmogorov–Smirnov test. **j** Hermaphrodites exposed to 2 nM of

medip#1 (red) ovulate earlier than controls (blue). Data represent one experiment with *n* = 25 animals for each condition. The differences at 54 h and 56 h are significant (*P* = 1.7 × 10⁻³ and *P* = 7 × 10⁻³, respectively, as calculated using binomial test). **k** Faster pharyngeal pumping in hermaphrodites treated with 2 nM of medip#1 (red) than in untreated controls (blue). Boxes represent the two inner quartiles, horizontal lines represent medians, and whiskers extend to 1.5× of the box data. Data represent one experiment with *n* = 30 animals for each condition; *P* value was calculated by Kolmogorov–Smirnov test. **l** Timepoint of first egg‑laying of isolated worms on different concentrations of medip#1 and nacq#1 compared to untreated isolated worms (ISO, control) and grouped worms (high density, HD). Data represent four biologically independent experiments, except for 10 µM medip#1 and 1 pM nacq#1 (one experiment), and 10 pM nacq#1 (three experiments), with the total number of animals used for each condition indicated above the *x* axis. *P* values were calculated by two‑sided Welch *t* test, comparing indicated conditions with ISO control. **m** Mean lifespan of medip#1‑treated WT animals compared to untreated control. Data represent three to four biologically independent experiments, each using 15–20 animals per plate; the total number of plates used for each condition is indicated above the *x* axis. *P* values were calculated by two‑sided *t* test, comparing indicated conditions with untreated control. Source data are provided as a Source Data file.

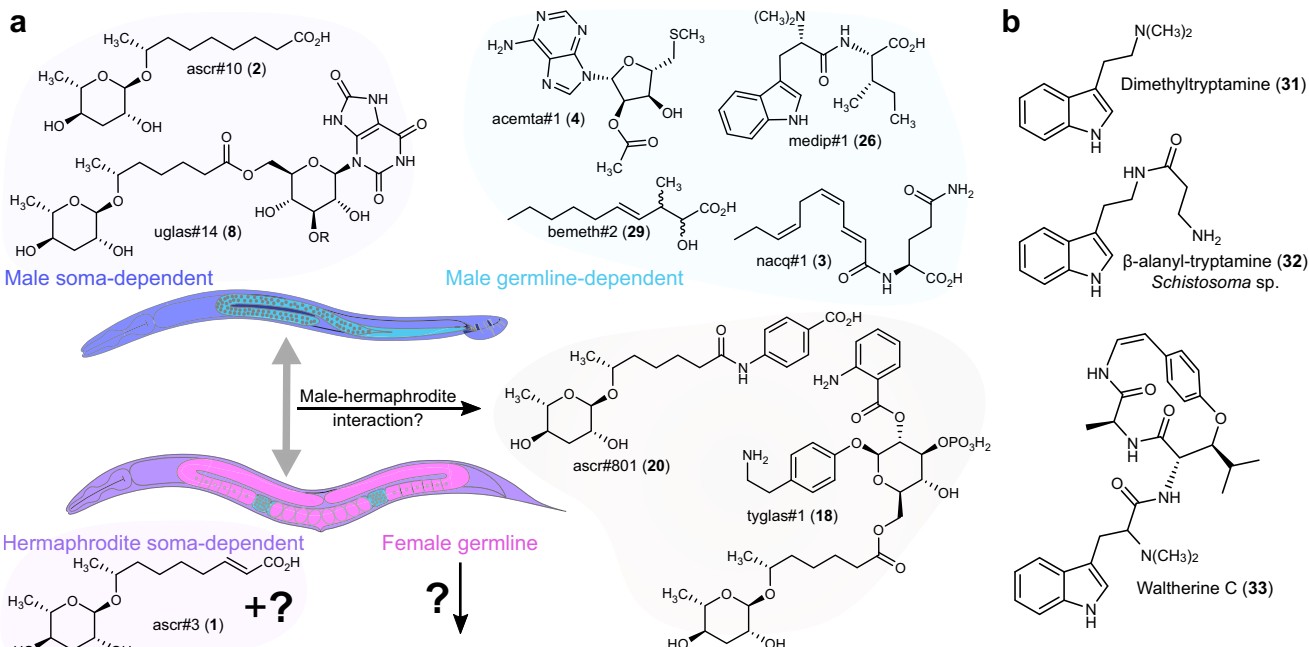

**Fig. 5 | Tissue-specific origin of sex-specific metabolites and related compounds from other natural sources. a** Soma- and germline dependence of selected male- and hermaphrodite-enriched metabolites. Reprinted from ref. [61] with permission from Elsevier. **b** Chemical structures of *N,N*-dimethyltryptamine, and *N,N*-dimethyltryptophan derivatives from other animals and plants.

characterization. It is likely that this inventory is still largely incomplete, as we used stringent peak intensity cutoffs for our analysis that may have excluded less abundant or less consistently produced metabolites. Furthermore, chromatographic conditions optimized specifically for the detection of very polar and very nonpolar metabolites (e.g., lipids) will almost certainly reveal additional male-upregulated compounds. Full chemical characterization of a large number of newly detected metabolites presents a major challenge, since structure elucidation cannot yet be automated and still relies on supervised one-by-one approaches. For this study, we therefore selected representative, abundant members of several families of male-enriched compounds for detailed characterization, which uncovered a wide range of unusual chemical structures integrating building blocks from diverse metabolic pathways.

Our analysis revealed that the production of a subset of male-enriched metabolites was correlated with the presence and sexual identity of the germline (Fig. 5a). Biosynthesis of the dipeptides medip#1–3 and the acetylated MTA derivatives acemta#1/2 was abolished in germline-feminized *fem-2(lf)* and germline-less *glp-4* animals and substantially increased in germline-masculinized *fem-3(gf)* animals. Consistent with the requirement of a male germline for their production, medip#1–3 and acemta#1/2 were absent in developing worms until germline maturation during the young adult stage. Several additional families of male-enriched metabolites not discussed in the preceding sections were also associated with the presence of a male germline. This includes a stark increase in the production of very long-chain polyunsaturated fatty acids, e.g., tetracosapentaenoic acid (**30**, Supplementary Fig. 27), which may be derived from the marked upregulation of two fatty acid elongases, *elo-4* and *elo-7*, in males[16]. Additional male germline-dependent metabolites include the recently reported β-methyldecanoic acid derivatives, bemeth#2 (**29**) and bemeth#3[8], as well as a number of putative modular glucosides and other compounds whose structures have not yet been elucidated (Supplementary Data 1–3). The abundance of male germline-dependent compounds we were able to detect in the small, hand-picked samples (e.g., nacq#1 and medip#1) did not change in hermaphrodites with the masculinized intestine, further supporting the

idea that the sex of the germline, rather than that of the soma, was the key determinant of their production.

In contrast, production of several other families of male-enriched metabolites does not depend on the presence of a male germline. For example, all male-enriched uric acid glucosides (e.g., uglas#14 and uglas#104) can still be detected in germline-feminized *fem-2(lf)* and germline-less *glp-4* animals. Notably, production of uglas#14 and uglas#104, which are more than tenfold enriched in WT or *him-5* males relative to WT hermaphrodites, was only slightly increased in germline-masculinized *fem-3(gf)* animals and did not differ significantly between *fem-3(gf)* and germline-less *glp-4* animals. *fem-3(gf)* animals have a largely hermaphrodite-like soma, indicating that upregulation of uglas#-family metabolites requires a male soma (Fig. 5a). Our finding that masculinization of the hermaphrodite intestine is sufficient to masculinize the profile of uglas#-family metabolites, as well as the ratio of the ascarosides ascr#3 and ascr#10, indicates that sexual specialization of metabolic pathways in the intestine is a key determinant for a subset of sexually dimorphic features of the metabolome.

A distinct pattern of occurrence is exemplified by the diverse family of *him-5*-enriched ascr#1 derivatives (Fig. 3). Although production of these compounds was dramatically increased in either the *exo-* or *endo*-metabolomes of male-enriched *him-5* cultures, most could not be detected in samples of pure males, except for ascr#1, which was not upregulated. Thus, increased levels of ascr#1 and ascr#1-incorporating modular metabolites (Fig. 3c) were observed only in cultures containing both males and hermaphrodites, but not in samples of pure WT or *him-5* males, suggesting that increased production of these compounds may result from interactions between the two sexes (Fig. 5a).

Given their elaborate structures and sex-specific production, it seems likely that many of the identified metabolites serve dedicated biological functions, e.g., as signaling molecules. Previously reported male-excreted compounds in *C. elegans* affect aspects of behavior[22,24], germline biology[26–28], and the rate of larval development[27,29]. Among the newly identified metabolites we selected the dipeptide medip#1 for further study, because its male-enriched production is conserved in *C. briggsae* and since it is abundantly produced in a male

germline-dependent manner. In addition, we were intrigued by the similarity of its chemical structure to that of other potently active biomolecules, e.g., dimethyltryptamine[49,50] (**31**, Fig. [5]b). We found that at physiologically relevant concentrations medip#1 accelerates larval development of hermaphrodites by shortening the duration of the last larval stage. Another male-enriched compound, nacq#1, similarly accelerates the last larval stage[29], consistent with the earlier finding that samples of the entire male *exo*-metabolome have the same effect[27]. The effects of nacq#1 and medip#1, despite their dissimilar chemical structures, appear to be redundant, since we did not observe additive or synergistic activity. Whether these two compounds accelerate development via a shared mechanism remains to be elucidated.

Both somatic and germline effects of medip#1 promote the male reproductive strategy of facilitating sexual maturation of potential mating partners. Similar observations in other species, including mammals[51], suggest that this may be a universal feature of social communication in animals. Interestingly, a dipeptide-like compound, β-alanyl-tryptamine (**32**, Fig. [5]b), was recently identified from male *Schistosoma mansoni* and, as medip#1, shown to stimulate female development[52]. Like schistosomal β-alanyl-tryptamine, medip#1 could be derived from a non-ribosomal peptide synthetase (NRPS). However, the *C. elegans* genome features only one NRPS-like gene, which has been shown to be involved in the biosynthesis of a different metabolite[53]. Moreover, the finding that dietary dimethyltryptophan is not incorporated into medip#1 may suggest a ribosomal origin. Cyclopeptides of likely ribosomal origin that include dimethyl-tryptophan have been described from plants, e.g., waltherine (**33**)[54]. Lastly, the abundant production of medip#1 and the identification of the likely SAM-derived acemta#1/2 may point to sex-specific differences in one-carbon metabolism.

Our results demonstrate that the male identity of the germline and the soma can regulate production of distinct sets of metabolites. Similarly, it can be expected that the hermaphrodite germline and soma contribute metabolites that are not, or only to a lesser extent, produced in males, and data deposited with this study provide a starting point for their analysis. It should be noted that correlation of a metabolite with the presence of the male or female germline may not necessarily imply that biosynthesis of that metabolite occurs, entirely or in part, within the germline itself. Alternatively, germline-dependent metabolites may result from activation or priming of somatic biosynthetic pathways by signals from the germline. Identification of genes involved in the biosyntheses of the different germline-dependent compound families will be an important next step toward uncovering their tissue origin. Taken together, our work highlights the power of untargeted comparative metabolomics in a genetically tractable model system to decipher the role of sex and the contributions of signals from different organ systems in shaping animal metabolomes.

## Methods

### Nematodes and bacterial strains
Unless otherwise indicated, worms were maintained at 20 °C on Nematode Growth Medium (NGM) petri dish plates seeded with *E. coli* OP50 obtained from the *Caenorhabditis* Genetics Center (CGC)[55]. Populations were synchronized by alkaline hypochlorite treatment of gravid hermaphrodites. Isolated eggs were rocked overnight in M9 buffer at 20 °C to yield synchronized, starved L1 larvae[56]. For animals that were sensitive to alkaline hypochlorite treatment, mixed-stage animals were grown at 15 °C and then washed off the plate with 20 mL M9 buffer which was collected in 50 mL conical tubes. Tubes were left for 10 min at room temperature, after which all non-L1 stages settled at the bottom of the tube. The top 15 mL, containing L1 animals, was transferred to a new conical tube, rinsed twice with M9 buffer, and then rocked overnight in M9 buffer at the nonpermissive temperature (25 °C) prior to experiments. The following strains were used for metabolomics experiments: wild-type Bristol N2, CB4088 *him-*

*5(e1490)*, DH245 *fem-2(b245)*, JK816 *fem-3(q20)* (gain-of-function), CB3844 *fem-3(e2006)* (loss-of-function), SS104 *glp-4(bn2)*, CB4037 *glp-1(e2141)*, JK569 *mog-3(q74)* PS7922 *cest-5.1(syb1131)*, FCS51 *cest-5.1(syb1131);him-5(e1490)*, PHX3933 *cest-5.1(syb1131);cest-5.2(syb3933)*, GR1395 *mgIs49 [mlt-10::GFP-pest;ttx-1::GFP]*, UR936 *fsEx445[Pnhx-2::FEM-3(+)::SL2::mCherry::unc-54_3'UTR;Psulp-3::gfp]*.

### *C. elegans* liquid cultures
Animals were grown at a density of three worms/1 μL at the indicated temperatures in S-Complete media supplemented with concentrated OP50. When animals reached adulthood, the cultures were settled as described above to separate adults from any offspring. The adult animals were rinsed three times with M9 and once with water to remove residual OP50 and salts. The supernatant containing L1 larvae was centrifuged at 1000×*g* for 1 min to remove L1 larvae and residual OP50. Adult worm pellets and conditioned media samples were snap-frozen and stored at −20 °C until extraction.

### 50:50 male:hermaphrodite samples
Four N2 animals were placed on 3.5-cm NGM plates seeded with OP50 and grown at 20 °C to the L4 stage. The worms were then incubated at 30 °C for 5 h, then transitioned back to 20 °C and allowed to reproduce. Males resulting from heat shock were then picked to new plates to mate with N2 hermaphrodites to yield populations of ~50% males.

### *C. elegans* plate-based cultures
Mixed-stage animals were grown on 10-cm NGM plates at 20 °C until nearly all of the food was depleted, at which point animals were rinsed off the plate with M9 and settled as described above. Isolated L1 larvae were transferred to new 10-cm NGM plates seeded with OP50. Worms were grown at 20 °C for 72 h to yield a synchronized population of 1-day-old adults, which were collected by washing the plate with M9. The animals were centrifuged at 1000×*g* for 1 min and the supernatant was transferred to a new tube, snap-frozen, and stored at −80 °C. The worms were rinsed twice with M9 and once with water, then frozen and stored at −80 °C until extraction.

### Hand-picked WT males and hermaphrodites
N2 and *him-5* animals were grown on 10-cm NGM plates seeded with OP50 as described above. After 72 h, 2000 N2 hermaphrodites, 2000 N2 males, and 2000 *him-5* males were picked with a platinum-tipped worm pick, rinsed twice with M9 and once with water, pelleted, frozen, and stored at −80 °C until extraction.

### Intestinally masculinized hermaphrodites
Within a 2-h window, for WT and intestinal masculinized UR936 animals, 250 adult day-1 hermaphrodites were hand-picked into 1 mL of OP50-containing M9 buffer (mCherry positive individuals were selected under a fluorescent volumetric microscope for the intestinal masculinized strain UR936). Then the vial was gently rocked at 20 °C for 5 h before the animals were separated from the supernatant by letting them settle to the bottom of the vial. Clear supernatant was transferred into an empty vial and residual worm pellet and supernatant were immediately frozen at −78 °C. A total of 20 vials (5000 animals) were collected per genotype. M9 buffer made from the same batch was used as the blank control in later chromatography experiments. To prepare the OP50-containing M9 buffer mentioned above, 3–6 mg of *E. coli* OP50 was collected from fresh liquid culture by centrifuge, LB media was thoroughly removed, and M9 buffer was added as the solvent to resuspend bacteria at 200 μL buffer per 1 mg bacteria.

### Developmental and adult time-course metabolomes
Approximately 100,000 synchronized L1 larvae obtained by alkaline hypochlorite treatment (described above) were grown at a

concentration of 3 worms/1 μL in S-Complete medium supplemented with concentrated OP50 at 20 °C. After 38 h when the majority of worms were in the 4th larval stage (L4), the cultures were transferred to 50 mL conical tube and allowed to settle for 10 min, as described above. The top 25 mL of the culture medium was transferred to a new conical tube, centrifuged at 1000×*g* for 1 min, and then snap-frozen. The animals were rinsed twice with M9 buffer, once with S-Complete buffer, and then resuspended at the same concentration in S-Complete medium and supplemented with OP50. The same protocol was followed, and the conditioned medium was harvested again at 52 h (worms in young adult stage), at 72 h (gravid adult stage), 96 h (day-1 adults), 120 h (day 2 adults), and 144 h (day 3 adults). The settling protocol was essential to separate newly hatched larvae from the aging adults; many larvae were present in the culture beginning at the 72-h timepoint. At the final harvest, the aged adults were pelleted, rinsed three times with M9 buffer and once with water, and then snap-frozen. Developmental profiling of *him-5(e1490)* cultures was performed using the same protocol but extended by 2 days. Conditioned medium was collected at 38, 52, 72, 96, 120, 144, 168, and 192 h after the experiment started, and the aged adults were collected at the final time as described.

## Procedure for liquid growth

N2 and *him-5* L1 animals were synchronized, and 100,000 animals were grown in S-Complete at 25 °C as described above. After 56 h the culture was settled, the pellet was rinsed three times with M9 and once with water, the supernatant was centrifuged to remove L1s and OP50, and both pellet and supernatant were frozen at −80 °C.

## Procedure for D$_3$-methyl-methionine feeding

Synchronized *him-5(e1490)* L1 larvae were prepared by alkaline hypochlorite treatment as described, and five separate cultures of 70,000 animals each were grown in S-Complete media supplemented with concentrated OP50. Cultures were grown without addition or were supplemented with L-methionine (Sigma-Aldrich M9625) or L-D$_3$-methyl-methionine (Cambridge Isotope Laboratories DLM-431) to achieve a final concentration of 10 mM. Two isotopic labeling experiments were performed in which animals were either supplemented beginning at the L4 stage (after 48 h in culture) or at the beginning of the experiment.

## Extraction procedure

Frozen worm pellets (*endo*-metabolome) and conditioned media (*exo*-metabolome) were lyophilized, and pellets were homogenized with a tissue grinder. Media and pellets were extracted in 30 mL and 13 mL pure methanol, respectively, for 16 h with shaking. The methanol extracts were separated from insoluble material by centrifugation, dried in vacuo, and resuspended in 50 μL of methanol for plated cultures and 150 μL of methanol for liquid cultures.

## Lipid-extraction procedure

Frozen pellets were lyophilized and homogenized as above, then extracted with 12 mL of a 9:1 ethyl acetate:ethanol mixture for 16 h with stirring. The organic extracts were treated as the methanol extracts above and resuspended in 150 μL of ethanol for analysis by HPLC-MS.

## UHPLC-HRMS

Liquid chromatography was performed using a Vanquish Horizon UHPLC controlled by Chromeleon software (ThermoFisher Scientific) coupled to an Orbitrap Q-Exactive HF high-resolution mass spectrometer controlled by Xcalibur software (ThermoFisher Scientific) or a Dionex Ultimate 3000 UHPLC coupled to an Orbitrap Q-Exactive high-resolution mass spectrometer controlled by the same software. UHPLC separation was achieved using a Thermo Hypersil GOLD C18 column

(2.1 × 150 mm 1.9 μm particle size) maintained at 40 °C. Solvent A: 0.1% formic acid in water; solvent B: 0.1% formic acid in acetonitrile. A/B gradient started at 1% B for 3 min after injection and increased linearly to 98% B at 20 min, followed by 5 min at 98% B, then back to 1% B over 0.1 min and finally held at 1% B for an additional 2.9 min.

For lipid extracts, reversed-phase post-column ion-pairing chromatography was performed using a Vanquish Horizon UHPLC coupled to an Orbitrap Q-Exactive HF as above. Solvent A: 0.1% ammonium acetate in water; solvent B: acetonitrile. A/B gradient started at 5% B for 3 min after injection and increased linearly to 98% B at 20 min, followed by 5 min at 98% B, then back to 5% B over 0.1 min and finally held at 5% B for an additional 2.9 min. A second pump (Dionex 3000) controlling a solution of 800 mM ammonia in methanol was run at a constant flow rate of 0.015 mL/min for the duration of the method and mixed via a micro-splitter valve (Idex #P-460S) with the eluate line from the column.

Mass spectrometer parameters: spray voltage (−3.0 kV, +3.5 kV), capillary temperature 380 °C, probe heater temperature 400 °C; sheath, auxiliary, and sweep gas 60, 20, and 2 AU, respectively. S-Lens RF level: 50, resolution 120,000 at *m/z* 200, AGC target 3E6. Each sample was analyzed in negative and positive electrospray ionization modes. Parameters for MS/MS (dd-MS2): MS1 resolution: 60,000, AGC Target: 1E6. MS/MS resolution: 30,000, AGC Target: 2E5, maximum injection time: 50 ms, isolation window 1.0 *m/z*, stepped normalized collision energy (NCE) 10, 30; dynamic exclusion: 5 s, top 10 masses selected for MS/MS per scan.

## Metaboseek analysis

UHPLC-HRMS data were analyzed using Metaboseek software after file conversion to the mzXML format via MSConvert (v3.0, ProteoWizard)[1,2]. A total of 41,795 features were detected in ESI- and 68,389 features in ESI +. No feature matching (ESI + with ESI−) was performed at this stage, and all features, including isotopes, adducts and fragments, were retained for the subsequent comparative analysis in Metaboseek. For subsequent prioritization of features, the following filtration criteria were used: (i) a minimum twofold increase in strain of interest over control (i.e., *him-5* over N2), (ii) a minimum average intensity of 100,000 arbitrary units for the feature in the strain of interest, and (iii) a *P* value less than 0.05 as calculated by two-sided, unpaired *t* test. MS2 networking was performed in MetaboSeek (version 0.9.7), Features were matched with their respective MS2 scan within an *m/z* window of 5 ppm and a retention time window of 15 s, using the MS2scans function. To construct a molecular network, the tolerance of the fragment peaks was set to an *m/z* of 5 ppm, the minimum number of peaks was set to 3, and the noise level was set to 2%. Once the network was constructed, a cosine value of 0.6 was used, the number of possible connections was restricted to 10. Molecular formulas were determined following comparative analysis via Metaboseek and MS2 networking, using the built-in MF-prediction tools of Thermo Xcalibur and Chemcalc (https://www.chemcalc.org/).

## Manual integration and normalization

For compounds discussed, the area under the curve (AUC) of the compound of interest was manually obtained through integration in Thermo FreeStyle and Qual Browser. Normalization to wild type (N2) was achieved by dividing the AUC of the compound of interest by the AUC of ascr#3, as detected in the *exo*-metabolome in ESI- ionization. The *P* values for metabolomics data were calculated via unpaired *t* tests with Welch correction.

## Quantification of medip#1 in *C. elegans endo*-metabolome samples

A concentration curve for medip#1 was generated by preparing a series of dilutions of synthetic medip#1 (35.7 nM, 357 nM, 3.57 μM), which were subsequently analyzed by ESI + MS analysis

along with *C. elegans* WT *endo*-metabolomic samples. Area under the curve for medip#1 ($m/z = 346.21251$) was determined using Fisher Scientific Freestyle software. Concentrations in analyzed WT (hermaphrodites) and *him-5* (~30% males) liquid culture *exo*-metabolome samples (volume 150 μL, see "Extraction Procedures" above) were determined as ~300–500 nM (WT, young adult stage) and ~1.6–2 μM (*him-5*, day-1 adults), corresponding to concentrations in the original liquid cultures (volume 25 mL) of roughly 2–3 nM (WT) and 10–12 nM (*him-5*), respectively. These estimates are likely below actual concentrations due to some compound loss during extraction.

### Conditioning plates with medip#1 and other compounds
Plates were conditioned as described in this paragraph, except for plates used in aging assays and the assay measuring the timepoint of first egg-laying, which are described separately in subsequent sections. Concentrated medip#1 was stored in ethanol at −20 °C. Dilutions were made in ethanol. For most treatments, 2 μM medip#1 solution was added to an equal volume of 1:10 dilution of OP50 overnight culture so that 1 μM medip#1 in a total volume of 20 μL was pipetted on to a 6-cm plate. This was quickly absorbed into the plate, and a second 20 μL drop of 1:10 dilution OP50 was pipetted over the first drop. Overnight diffusion of the added compound throughout the total volume of agar in the plates (10 mL) is expected to yield a final assay concentration of ~2 nM. For experiments testing medip#1 at different concentrations, stock solution was adjusted accordingly to yield the desired final concentration. Conditioning of plates with other compounds was done analogously. Control plates were made with ethanol alone added to OP50 bacteria as above. These plates were incubated overnight at 20 °C and used the following day.

### Acceleration of reaching adult morphology
Acceleration of reaching adult developmental morphology (Supplementary Fig. 24) was determined using the protocol described in ref. [27]. Briefly, after synchronization by hypochlorite treatment, 25 L1 larvae were singled onto control plates and 25 L1 larvae were singled onto plates prepared with 2 nM medip#1 as above. Beginning at 48 h post release from L1 arrest, worms were monitored every 2 h for transition from L4 to young adult. The developmental staging was based on vulval and gonadal morphology[57]. These results were used to calculate the difference in time of development when half the population had reached adulthood between worms exposed to 2 nM medip#1 and their paired controls.

### Census of the reproductive system
A census of oocytes in the gonad and embryos in the uterus was performed using an established protocol[58]. Approximately 30 synchronized L1 larvae prepared by alkaline hypochlorite treatment were pipetted onto either control NGM plates or NGM plates prepared with 2 nM medip#1, as described above. Every 2 h, beginning at 48 h release from post L1 arrest, 25 worms from each condition were examined on a Leica DM5000B compound microscope. The number of oocytes that completely spanned the gonad in both the anterior and posterior arms, as well as the number of fertilized embryos in the uterus, were counted. In addition, the fraction of worms that had ovulated at least once was noted.

### Pharyngeal pumping rate
N2 hermaphrodites were synchronized by alkaline hypochlorite treatment and reared on control NGM plates for 72 h. At that time, ten hermaphrodites were transferred to either control NGM plates or 2 nM medip#1 plates prepared as above, and then allowed to acclimate for an hour. After acclimation, individual animals were monitored and the number of pharyngeal pumps occurring in 20 s was counted. This was done three times for each individual, waiting at least 20 s between counts. The pharyngeal pumping rates for 30 worms were measured for 2 nM medip#1 and control.

### Generation of *mlt-10* molting curves
The molting curves were generated using the protocol described in ref. [59], except that the timing of the experiment was limited to 22 to 36 h post release from L1 arrest, covering L1 through L3 larval stages. The experiment relied on monitoring the level of GFP in GR1395 *mgIs49 [mlt-10::GFP-pest; ttx-1::GFP]* hermaphrodite larvae[60]. Worms were maintained at 20 °C on OP50 *E. coli* under standard nematode growth conditions[55]. Populations were synchronized by alkaline hypochlorite treatment of gravid hermaphrodites. Isolated eggs were allowed to hatch overnight in M9 buffer with rotation at 20 °C[56]. Between 50 and 60 L1 larvae from this population were transferred to either control plates or 2 nM medip#1 plates. Each of the two experiments consisted of six control and six treatment plates, for a total of 678 hermaphrodites (342 controls and 336 2 nM medip#1). The identity of the plates was not known to the experimenter. Animals were examined every hour and scored for GFP fluorescence on a Leica MZ16F stereomicroscope.

### Developmental acceleration/timepoint of first egg-laying
Synchronized fertile adults (60 animals) were allowed to lay eggs for 2 h on 6-cm NGM plates coated with OP50 *E. coli* bacteria. Resulting individual eggs (ISO = isolated condition) or 40 eggs per plate (HD = high density condition) were transferred onto 3.5-cm NGM plates (volume: 4 mL of agar) that had been prepared as follows: For assays with nacq#1 or medip#1, 100 μL of aqueous stock solutions of medip#1 (400 μM, 40 μM, 4 μM, 400 nM, or 40 nM) and nacq#1 (40 nM, 4 nM, 400 pM, or 40 pM) were distributed on the plate surface, resulting in plates containing 10 μM, 1 μM, 100 nM, 10 nM, or 1 nM of medip#1, plates containing 1 nM, 100 pM, 10 pM, or 1 pM of nacq#1, as well as mock-treated (control) plates. Analogously, plates containing 10 nM of medip#1 plus 10 pM nacq#1 as well as plates containing 100 nM or 10 nM, of Trp-Ile or Ile-Trp were prepared analogously. Plates were dried for 20 min with open lids in a fume hood. Then 30 μL of *E. coli* bacteria grown overnight in LB solution were placed in the middle of each plate. Plates were dried for 20 min and incubated overnight at room temperature before use. As the start time of the assay, the timepoint of egg transfer was defined. At 59 h at 20 °C, 7–15 individual worms grown under ISO and HD conditions on the compound or mock-treated plates were transferred onto fresh compound or mock-treated plates. At 60 h and then every hour, worms were scored for egg lay. Animals were scored using a Leica DM 5500B microscope.

### Lifespan assays
Synchronized fertile adults (60 animals) were allowed to lay eggs for 2 h on 6-cm NGM plates seeded with OP50 *E. coli* bacteria. Subsequently, eggs were transferred to 3.5-cm NGM plates (20 eggs per plate, 4 mL plate volume) that were treated as follows: 100 μL of aqueous stock solutions of medip#1 (400 μM, 40 μM, 4 μM, or 400 nM) or solvent control (0.1% ethanol in the added volume of 100 μL) were distributed on the plate surface, resulting in plates containing 10 μM, 1 μM, 100 nM, or 10 nM of medip#1, and mock-treated plates (control). Plates were dried for 20 min under a fume hood. 30 μL of *E. coli* bacteria grown overnight in LB solution were placed in the middle of each plate. Plates were dried for 20 min and incubated overnight at room temperature before use in the experiment. From day 3 on, worms were transferred to fresh experimental plates daily, after 8 days, when animals ceased to lay eggs, they were transferred every 3 days to

fresh plates. Worms were checked daily for survival, and dead animals were removed. Animals were scored using a Leica DM 5500B microscope.

## Data and statistical analysis
LC-MS data were collected using Thermo Scientific Xcalibur software version 4.1.31.9. LC-MS data were analyzed using Metaboseek software version 0.9.7 and quantification was performed via integration in Excalibur Quan Browser version 4.1.31.9. NMR spectra were processed and baseline corrected using MestreLabs MNOVA software packages version 11.0.0-17609. All statistical analyses were performed with GraphPad Prism (versions 9.2 and 9.4.1), Metaboseek (version 0.9.7) or R (version 4.1.1). The $P$ values of datasets were determined by two-tailed Welch $t$ tests, unless specified otherwise. Error bars represent standard errors of the mean (SEM).

## Reporting summary
Further information on research design is available in the Nature Portfolio Reporting Summary linked to this article.

## Data availability
MS and MS/MS data are available at GNPS/MassIVE under accession code MSV000089965. See MS Data Inventory (Supplementary Data 4) for file names and sample identities. Source data are provided with this paper.

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

## Acknowledgements

This research was supported in part by the National Institutes of Health (R35GM131877 and U2CES030167 to FCS, and NIH R01 GM140415 to D.S.P.) and the Howard Hughes Medical Institute (Faculty Scholar grant to F.C.S.). E.Z.A. and I.R. were supported in part by an NSF grant (IOS-1755244) to I.R. Some strains used in this work were provided by the CGC, which is funded by the NIH Office of Research Infrastructure Programs (P40 OD010440). We thank Gary Horvath for technical support and Ivan Keresztes and David Kiemle for assistance with NMR spectroscopy.

## Author contributions

F.C.S. and I.R. supervised the study. R.N.B., A.B.A., B.W.F., S.S.L., I.R., D.S.P., and F.C.S. designed the study. R.N.B., A.B.A., E.Z.A., B.W.F., B.J.C., A.H.L., O.P., C.J.W., J.L., D.F.P., V.B., and A.C. performed chemical and biological experiments. R.N.B., I.R., and F.C.S. wrote the paper with input from all authors.

## Competing interests

F.C.S. is a cofounder of Ascribe Bioscience Inc. and of Holoclara Inc. The remaining authors declare no competing interests.
