## [Peer Review File · Nature Communications]

Sex-specificity of the *C. elegans* metabolomeREVIEWER COMMENTS

Reviewer #1 (Remarks to the Author):

C. elegans is arguably one of the most important and well-studied of invertebrate model systems yet the roles that small molecules play in animal to animal communication remains an understudied area of investigation. In this study Burkhardt et al. generate a highly detailed catalog of small molecule metabolites produced by male worms. The work not only confirms the existence of previously identified molecules, it uncovers an array of novel metabolites present both endogenously and released to environment from male worms. Among the new molecules uncovered is a novel dipeptide-like molecule that can accelerate the development of hermaphrodite worms. Overall, the manuscript is well-written and describes the most comprehensive analysis of the male *C. elegans* metabolome to date. As such, I anticipate the work will be of interest to those in the field as well as those outside the field of *C. elegans* biology. I have only minor comments/suggestions for clarification:

1. It was unclear how the concentration of medip#1 was chosen for assay with hermaphrodites. Although it is mentioned in the Discussion that levels used were "physiologically relevant", how was this determined? Is this inferred from the MS data? Also, were dose response studies conducted and at what concentrations were effects observed?
2. It is interesting that both medip#1 and nacq#1 have similar effects accelerating development. Could this be due to a non-specific response to xenobiotics? Do medip#1 analogs (e.g., medip#2-3) have any effects? This experiment could help support the model this is a specific effect of medip#1.
3. Related to point 2 above. Have the authors evaluated whether the effects of medip#1 and nacq#1 synergistic with one another (i.e., do they accelerate development at sub-optimal concentrations when co-administered)? This could provide insight into the mechanism of by which these molecules function.

Minor comments

Line 175: Change "primary" to "primarily"

Reviewer #2 (Remarks to the Author):

The article by Burkhardt et al. deals with the sex-specific metabolome of the nematode *Caenorhabditis elegans*. In order to enrich males, which are only observed at lower than <1% under standard conditions of the otherwise hermaphroditic worm, different mutants as well as hand-picked males have been used. On these samples non-targeted metabolomics using high-resolution mass spectrometry. The methodology is mostly scientific sound, but several minor points need to be clarified. Otherwise, the manuscript is well-written and easy to follow.

- On page 5 the authors stated that they have detected 110184 features. I'm assuming these are not isotope grouped since in the next sentence manual curation to remove isotopes, adducts and fragments has been mentioned. Why was no automatic grouping such as CAMERE, cliqueMS or similar used? This would give an estimate of metabolites detected.
 - How were molecular formulas calculated? Was this done manually or automatically? Which tools has been used?
 - In the SI Table 1 it is not indicated if the compounds were detected in positive or negative ionization mode. Likewise, the main text does not describe if the 110184 detected features are found in both ionization modes or only in one ionization mode? In the latter case, in which one?
 - How was the MS2 networking performed? GNPS? Was it the classic GNPS workflow or Feature-based-molecular networking? Please add the parameters that have been used for the MS2 networking.
 - Will the newly described metabolites be available via SMID-DB together with their MS2-spectra?
- Beside these technical questions a few questions related to the studied biology remain
- It is interesting that the authors find derivatives of a degradation product of SAM as well as a di-

methylated peptide. This would somehow show sex-specific differences in one-carbon metabolism. Is something know about that?

- Did the authors check for further demethylated peptides? Do the authors think that the methylation is specific to the dipeptide they found or a general phenomenon?

- Were additional controls used for the developmental acceleration? Was the non-methylated dipeptide tested as well as control?

Resubmission of *Nature Communications* manuscript NCOMMS-22-29597:

Burkhardt *et al.*

Sex-specificity of the *C. elegans* metabolome

We thank the Reviewers for their insightful and constructive comments on the above manuscript and provide a detailed point-by-point response below.

Reviewer #1:

C. elegans is arguably one of the most important and well-studied of invertebrate model systems yet the roles that small molecules play in animal to animal communication remains an understudied area of investigation. In this study Burkhardt et al. generate a highly detailed catalog of small molecule metabolites produced by male worms. The work not only confirms the existence of previously identified molecules, it uncovers an array of novel metabolites present both endogenously and released to environment from male worms. Among the new molecules uncovered is a novel dipeptide-like molecule that can accelerate the development of hermaphrodite worms. Overall, the manuscript is well-written and describes the most comprehensive analysis of the male C. elegans metabolome to date. As such, I anticipate the work will be of interest to those in the field as well as those outside the field of C. elegans biology. I have only minor comments/suggestions for clarification:

We thank this Reviewer for their time, and we are very grateful for the valuable suggestions, which we address in detail below:

1. It was unclear how the concentration of medip#1 was chosen for assay with hermaphrodites. Although it is mentioned in the Discussion that levels used were “physiologically relevant”, how was this determined? Is this inferred from the MS data? Also, were dose response studies conducted and at what concentrations were effects observed?

We agree that this could be clarified better. Therefore, we added a paragraph describing calculation for media concentrations of medip#1 in the WT and *him-5* exo-metabolomes in the Methods section. The results - 2-3 nM in WT and 10-12 nM in *him-5* - served as an initial guide for selecting assay concentrations.

2. It is interesting that both medip#1 and nacq#1 have similar effects accelerating development. Could this be due to a non-specific response to xenobiotics? Do medip#1 analogs (e.g., medip#2-3) have any effects? This experiment could help support the model this is a specific effect of medip#1.

We thank this reviewer for this important question, which was also raised by reviewer#2. In this revision, we added extensive data that demonstrate specificity of the activity of medip#1:

- We show that, while the *C. elegans* endo-metabolome includes non-methylated Trp-Ile and Ile-Trp, their abundances are not significantly different between animals with a female germline (*fem-2* (*lf*)) and a masculinized germline (*fem-3* (*gf*)) (new Figure S1d).

- We then demonstrate that dipeptides closely related to medip#1 but lacking methylation, specifically Trp-Ile and the reverse combination, Ile-Trp, are inactive in the developmental acceleration and pharyngeal pumping assays (new Figures S25a-d).
- We tested another male-enriched metabolite that is structurally unrelated to medip#1, the hydroxylated branched-chain fatty acid bemeth#2, for acceleration of larval development and effects on pharyngeal pumping. bemeth#2 was inactive at all tested concentrations (Figure S25 e, f), further supporting the conclusion that the responses to medip#1 are structure-specific.
- Lastly, we added new data showing that medip#1 production is unchanged in hermaphrodites with a masculinized *intestine*, further supporting the specifically male germline-dependent production of this methylated dipeptide (new Figure S1b).

3. Related to point 2 above. Have the authors evaluated whether the effects of medip#1 and nacq#1 synergistic with one another (i.e., do they accelerate development at sub-optimal concentrations when co-administered)? This could provide insight into the mechanism of by which these molecules function.

Under laboratory conditions, the effects of nacq#1 or medip#1 are not additive or synergistic at the tested concentrations (new Figure S26); however, development is affected by a wide range of nutritional and environmental parameters and thus we cannot exclude that under more natural settings the two compounds may synergize or act additively. Toward a mechanistic understanding of the effects of medip#1 and nacq#1, identification of required sensory neurons will be an important step, in combination with mutant screens for receptors and downstream components of signaling cascades regulating development and feeding behavior. These are extensive studies that are beyond the scope of current paper, which will serve as a foundation for mechanistic studies of how the genetic sex of soma and germline shape *C. elegans* metabolism and small molecule-dependent phenotypes..

Minor comments

Line 175: Change “primary” to “primarily”

Done, thanks!

Reviewer #2:

*The article by Burkhardt et al. deals with the sex-specific metabolome of the nematode *Caenorhabditis elegans*. In order to enrich males, which are only observed at lower than <1% under standard conditions of the otherwise hermaphroditic worm, different mutants as well as hand-picked males have been used. On these samples non-targeted metabolomics using high-resolution mass spectrometry. The methodology is mostly scientific sound, but several minor points need to be clarified. Otherwise, the manuscript is well-written and easy to follow.*

We thank this reviewer for their excellent comments and suggestions, which we address below.

- On page 5 the authors stated that they have detected 110184 features. I'm assuming these are not isotope grouped since in the next sentence manual curation to remove isotopes,

adducts and fragments has been mentioned. Why was no automatic grouping such as CAMERE, cliqueMS or similar used? This would give an estimate of metabolites detected.

We initially used CAMERA (which is an option implemented in Metaboseek) for this dataset; however, we found that annotation of less abundant compounds was not consistent. Generally, we find it preferable to retain all features for untargeted comparative analyses that aim to discover a small subset of metabolites. In this scenario, the large number of features and the presence of adducts, isotopes etc. does not present a challenge, since the resulting list of differential features will be comparatively small and thus manageable for manual curation. We added a sentence in Methods (line 632) stating that all features, including isotopes, adducts and fragments, were retained for the comparative analysis in Metaboseek.

- How were molecular formulas calculated? Was this done manually or automatically? Which tools has been used?

Molecular formulas were determined based on manual analysis using the built-in MF-prediction tools of Thermo Xcalibur and Metaboseek/Chemcalc; then verified by comparing simulated and experimental high resolution MS1 isotope patterns, and finally further validated using MS2 spectra. We added a sentence in Methods for clarification (line 641).

- In the SI Table 1 it is not indicated if the compounds were detected in positive or negative ionization mode.

Thanks pointing this out; we added this information to Table S1.

Likewise, the main text does not describe if the 110184 detected features are found in both ionization modes or only in one ionization mode? In the latter case, in which one?

The given number refers to the total numbers of detected features from both ionization modes. We added numbers for detected features in ESI+ and ESI- in Methods.

- How was the MS2 networking performed? GNPS? Was it the classic GNPS workflow or Feature-based-molecular networking? Please add the parameters that have been used for the MS2 networking.

MS2 networking was performed in Metaboseek using the default parameters (5ppm/15s window, cosine 0.6, max. 10 connections). We added this information in Methods (see paragraph "Metaboseek Analysis").

- Will the newly described metabolites be available via SMID-DB together with their MS2-spectra?

Yes, all newly described metabolites and their MS2 spectra will be included on SMID-DB upon acceptance of the manuscript for publication. We added a sentence in data availability for clarification.

Beside these technical questions a few questions related to the studied biology remain

- It is interesting that the authors find derivatives of a degradation product of SAM as well as a dimethylated peptide. This would somehow show sex-specific differences in one-carbon metabolism. Is something known about that?

We agree; this is an interesting question. There is one annotated methyl transferase that is significantly more highly expressed in the male germline, F13D12.9 (recently named *fcmt-1*); however, in a separate study we have shown that *fcmt-1* mutants are not defective in the biosynthesis of *medip#1*. Nonetheless, we agree this is an interesting question and added a sentence in the discussion to highlight that our data, especially the identification of the SAM-related *acemta#1/2*, suggest sex-specific differences in one carbon metabolism.

- Did the authors check for further demethylated peptides? Do the authors think that the methylation is specific to the dipeptide they found or a general phenomenon?

We thank this reviewer for bringing up this important point. We did not find any other dimethylated peptides in the list of male-upregulated metabolites (Table S1). Specificity of *medip#1* biosynthesis is further supported by the absence of the leucine-incorporating isomer (instead of isoleucine) of *medip#1*, *N,N*-dimethyl-tryptophan-*leucine* (Figure 4c).

Further supporting specificity of *medip#1* production, we added new data showing that, while the *C. elegans endo*-metabolome also includes the non-methylated dipeptides Trp-Ile and Ile-Trp, their abundances are not significantly different between animals with a female germline (*fem-2* KO) and a masculinized germline (*fem-3* OE) (new Figure S1d).

In addition, we included new data showing that *medip#1* production is unchanged in hermaphrodites with a masculinized *intestine*, supporting the specifically male germline-dependent production of the methylated dipeptide *medip#1* (Figure S1). Analysis of intestinally masculinized hermaphrodites additionally demonstrated that production of male-specific ascaroside profiles and the male-specific *uglas*-family compounds is determined by the sex of the intestine, providing additional support for one of the main conclusions, that soma and germline contribute distinct sets of sex-specific metabolites. The new data for intestinally masculinized hermaphrodites are shown in Figures 2j, 3b, and S1.

- Were additional controls used for the developmental acceleration? Was the non-methylated dipeptide tested as well as control?

We thank this reviewer for this important question, which was also raised by reviewer#1. In this revision, we added data that demonstrate that dipeptides closely related to *medip#1* but lacking methylation, specifically Trp-Ile and the reverse combination, Ile-Trp, are inactive (new Figure S25).

REVIEWERS' COMMENTS

Reviewer #2 (Remarks to the Author):

The authors did a great job in answering all questions and concerns raised by both reviewers. Specific technical questions raised by me have been answered in detail. The work is suitable for publication, I just insist on making all the data available via Metabolights, Massive or similar repositories and submit the new molecules to SMID-DB.

Resubmission of *Nature Communications* manuscript NCOMMS-22-29597A

Burkhardt *et al.*

Sex-specificity of the *C. elegans* metabolome

Reviewer #2:

The authors did a great job in answering all questions and concerns raised by both reviewers. Specific technical questions raised by me have been answered in detail. The work is suitable for publication, I just insist on making all the data available via Metabolights, Massive or similar repositories and submit the new molecules to SMID-DB.

Response: All data have been made available at MASSIVE under accession number <ftp://massive.ucsd.edu/MSV000089965/> and all new molecules have been submitted to SMID-DB. We gain thank the reviewers for their excellent comments and suggestions.